# Lean-ing on Quality:
# How High-Quality Data Beats Diverse Multilingual Data in AutoFormalization

## Abstract

*Autoformalization*, the process of transforming informal mathematical language into formal specifications and proofs remains a difficult task for state-of-the-art (large) language models. Existing works point to competing explanations for the performance gap. On one hand, large language models exhibit exceptional performance on translation tasks, suggesting their significant potential for autoformalization. On the other hand, the quantitative reasoning capabilities of standard language models remain limited, leading to suboptimal performance on autoformalization and the subsequent task of formal theorem proving. To this end, we introduce a novel methodology that leverages backtranslation with hand-curated prompts to enhance the mathematical capabilities of language models, particularly addressing the challenge posed by the scarcity of labeled data. Specifically, we evaluate three primary variations of this strategy: (1) on-the-fly (online) backtranslation, (2) distilled (offline) backtranslation with few-shot amplification, and (3) line-by-line proof analysis integrated with proof state information. Each variant is designed to optimize data quality over quantity, focusing on the high fidelity of generated proofs rather than sheer data scale. Our findings provide evidence that employing our proposed approaches to generate synthetic data, which prioritizes quality over volume, improves the autoformalization performance of LLMs as measured by standard benchmarks such as ProofNet. Crucially, our approach outperforms pretrained models using a minimal number of tokens. We also show, through strategic prompting and backtranslation, that our approaches surpass the performance of finetuning with extensive multilingual datasets such as MMA on ProofNet with only 1/150th of the tokens. Taken together, our methods show a promising new approach to significantly reduce the resources required to formalize proofs, thereby accelerating AI for math.

## 1 Introduction

Neural machine translation has been a focal point of research since the early development of machine learning (Dong et al., 2015) (Wu et al., 2016). *Autoformalization* can be viewed as a specialized application of this approach, where the objective is to translate theorems from informal, human-readable markup languages like LaTeX into formal languages such as Lean4 — a programming language specifically designed to encode complex mathematical constructs (de Moura & Ullrich, 2021). In principle, the development of an agent capable of precise autoformalization would significantly reduce the prohibitive costs of manually formalizing proofs. This would have profound implications; such an advancement could render all mathematical knowledge, much of which is presently recorded in natural language, programmable: substantially enhancing the usability of interactive theorem proving systems and accelerating the expansion of human mathematical understanding (Klein et al., 2009).

Autoformalization first gained traction in 2018 when researchers used long short-term memory networks to generate statements in Mizar (Wang et al., 2018). More recently, experiments demonstrated promising results using a naive few-shot learning approach to translate English into Isabelle code (Wu et al., 2022). Historically, researchers have focused on merging automated theorem provers with language models, which often struggle with quantitative reasoning, limiting the types of theo-

rems that can be expressed and solved and reducing the models' effectiveness in complex, real-world scenarios. Our study adopts a novel approach by integrating interactive theorem provers with large language models, leveraging the sequential processing capabilities of both. By fine-tuning LLMs with intermediary statements from the LeanDojo dataset (Yang et al., 2023), we aim to enhance our understanding of formal theorem provers' reasoning processes and successfully incorporate this signal into an LLM.

While LLMs have performed well on benchmarks related to various natural language understanding tasks as evaluated by benchmarks such as MMLU (Hendrycks et al., 2020), they have continued to struggle with tasks that require deeper quantitative reasoning (Lewkowycz et al., 2022). Moreover, while strides have been made in using deep learning for symbolic mathematics (Lample & Charton, 2020), propositional logic (Hahn et al., 2021), and differential systems (Charton et al., 2021), the field of autoformalization and theorem proving has generally not seen proportional advances. This disparity can largely be attributed to two main factors:

1. **Complexity:** Current models exhibit a gap in their ability to translate intricate mathematical concepts from natural language to formal logic.

2. **Scarcity:** The available training data for fine-tuning models in formal languages like Lean is significantly limited. Given the specialized nature of these programming languages, the sum total of formal-math language data is a minuscule fraction of the size of modern LLMs' training datasets. Thus, except for hand-curated benchmarks such as ProofNet (Azerbayev et al., 2023a) and MiniF2F Zheng et al. (2021), there is virtually no paired formal-informal data.

---

**Example of Formal and Informal Statements**

**Informal:** If $r$ is rational ($r \neq 0$) and $x$ is irrational, prove that $r + x$ is irrational.
**Formal:** theorem exercise_1_1a (x : R) (y : Q) : ( irrational x ) $\rightarrow$ irrational ( x + y ) :=

**Informal:** In an additive group $G$, the subtraction of vectors $g_1$ and $g_2$ is equal to the subtraction of the group elements $g_1$ and $g_2$.
**Formal:** theorem vsub_eq_sub {G : Type*} [AddGroup G] (gž081 gž082 : G) : gž081 -Ĭd65 gž082 = gž081 - gž082 := rfl #align vsub_eq_sub vsub_eq_sub section General variable G : Type* P : Type* [AddGroup G] [T : AddTorsor G P]

---

Figure 1: **Autoformalization Task Example**. This table compares formal Lean statements with their equivalent informal mathematical statements. The formal statements represent theorems in Lean, while the informal statements convey the theorems in human-readable natural proof language.

We present a new autoformalization dataset, **AI4Math**, which pairs statements from natural language proofs with their corresponding translations in Lean, as shown in Figure 1. Our approach to generating this dataset leverages backtranslation: a method in neural machine translation for creating synthetic training data (Liu et al., 2021). Given a monolingual dataset in the target language, backtranslation proceeds as follows:

1. Utilize a pretrained model to generate translations in the reverse direction, i.e., from the target language to the source language.

2. Assemble a parallel corpus consisting of these synthetic source and ground-truth target pairs.

3. Fine-tune or train on the generated dataset, enabling the model to translate the synthetic source examples back to the target language.

Our dataset, **AI4Math**, serves two primary purposes: first, to train an autoformalizer on individual statement pairs, and second, to fine-tune an LLM on the state of the proof before and after a specific tactic is applied. This approach aims to closely mimic the reasoning process in theorem proving, enhancing the LLM's ability to understand and generate formal proofs.

Our **key contributions** are:

1. We introduce a new paradigm for generating synthetic formal-informal paired statements, emphasizing the superiority of high-quality data over large but undifferentiated datasets.

2. We advance the discussion on data efficacy in neural language model training by demonstrating that richer prompts and high-quality Lean statement selections yield better performance than training on large, randomly diverse data.

3. By integrating proof state matching and manually curating data to include the state before and after a proof statement, we achieve dramatic improvements in model performance. This method not only optimizes the training process but does so with a minimal number of tokens, thereby enhancing computational efficiency and model effectiveness. Our approach underlines the potential of targeted and intelligent data preparation in maximizing the value of each token in autoformalization datasets.

## 2 Training Data Generation via Backtranslation

Backtranslation has been a pivotal method in neural machine translation (NMT), with its demonstrated utility for enhancing training efficacy in the absence of sufficient labeled data (Poncelas et al., 2018). This technique was further refined in transformer models (Han et al., 2021) and in preliminary experiments involving distilled backtranslation within the domain of autoformalization (Azerbayev et al., 2023a). We chose this methodology because it is one of the most prominent and validated methods for NMT tasks (Liu et al., 2021), and has been shown to be analytically and empirically suitable for neural model training (Jiang et al., 2023).

Our approach is to start with data in formal language (FL) and translate it into informal language (IL). We refer to this process as *informalization*. We demonstrate that each of these three dataset generation methodologies can improve model performance:

- **On-The-Fly Backtranslation:** This strategy for data augmentation utilized a unique backtranslation approach integrated within a custom training loop. Specifically, the training loop itself has four main steps, as outlined in Figure 2: (1) translating a batch of FL examples to IL (2) translating the synthetic IL examples back to FL, using the (synthetic IL, FL) pairs as labeled training data (3) computing the loss of the generated FL translations compared to the original FL examples and (4) backpropagating to update the model weights.

  This method diverged from traditional practices by not employing separate teacher and student models; instead, it trained a single model to simultaneously manage both directions of the translation task. This setup was chosen because it enabled the model to iteratively generate its own training data at each step, and this self-sufficient nature allowed us to avoid the problem of limited labeled data without incurring large API costs.

  The relatively small size of the model used in our experiments, GPT-2 with 124M parameters, suggested that it would not be very effective at generating high-quality synthetic informalizations. Despite this, the on-the-fly backtranslation method led to a significant reduction in evaluation loss after just a few hundred training steps, eventually leading to a substantial improvement compared to the baseline as seen in Table 2. Nevertheless, we eventually reached a performance plateau due to the inherent limitations of the baseline model. Due to resource constraints, we were unable to validate this method with a larger, more capable model such as Llemma-7B (Azerbayev et al., 2023b). We hypothesize that employing a larger pretrained model could yield even more promising results. This method thus serves as a baseline for comparison, and can highlight the potential benefits of using more powerful models in future work.

- **Distilled Backtranslation:** Considering the limitations of our initial approach, we pivoted to a more robust technique known as *distilled backtranslation*. For this method, we utilized a more powerful pretrained model, GPT-4 (OpenAI, 2023), as the teacher model by feeding it a dataset of FL examples and utilizing it to generate corresponding synthetic IL text, as outlined in Figure 3. We capitalized on the strong in-context learning capabilities of LLMs (Brown et al., 2020) by employing a few-shot prompt that included six labeled examples of FL-to-IL translations. This few-shot amplification approach allowed us to develop a comprehensive labeled dataset for training, rather than generating data on-the-fly. The full technique and prompt can be found in 5, with a formal example in Appendix A.

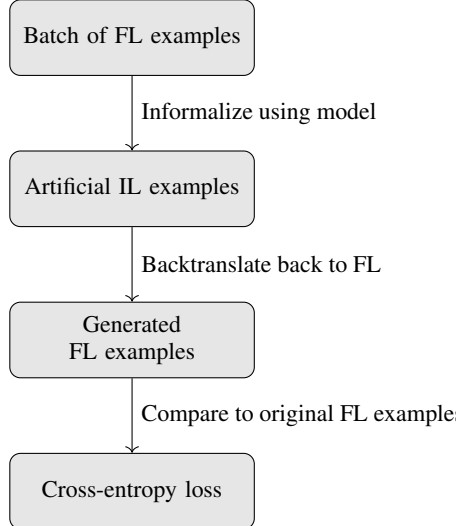

Figure 2: **On-The-Fly Backtranslation Process Graph**. Given the scarcity of pairs of formal and informal mathematics, our solution is to generate artificial informal language (IL) translations from a formal language dataset (FL), then use the (IL, FL) pairs for training. This methodology first uses the LLM to translate from FL to IL, and then translates back to FL. This avoids the problem of lack of paired data by having the model generate training data on the fly to teach itself.

Specifically, we extracted theorems from the MathLib4 dataset (mathlib Community, 2019) and then used GPT-4, enhanced by few-shot prompting, to informalize each theorem individually. Using regular expressions, we parsed over 100,000 theorems which we could later informalize and fine-tune on. However, due to the relatively high cost of using OpenAI's GPT-4 API, we were only able to informalize a small subset of these proofs as shown in Table 1. Within this scope of distilled backtranslation, we compared several ways of looking at proofs:

1. Our first method, inspired by techniques in Azerbayev et al. (2023a), involves informalizing entire theorems while using the full proof as contextual support in the prompt, generating the theorem statement in natural language. We employed a 6-shot prompt to create a dataset of 348 training pairs for this approach.

2. Our second method focuses on informalizing not only the theorem statement but also each individual tactic within a proof. This approach tests whether detailed explanations of each tactic can enhance the overall understanding of the proof. Utilizing the LeanDojo dataset Yang et al. (2023), which is pre-parsed into distinct tactics, our prompts included the formal theorem statement and the states of the proof both before and after each tactic is applied. We employed a zero-shot prompting strategy with tuples of (stateBefore, tactic, stateAfter) to generate a dataset of individually translated tactics, aiming to improve translations between the complete Lean theorem and its natural language counterpart.

• **Gathering Data from Regular Expressions:** We also prepared a large dataset leveraging only regular expressions for capturing specific lines/keywords in tactic scripts. Utilizing regular expressions for parsing LeanDojo proof tactics allowed us precise control over the selection of specific tactic lines and proof methodologies, enabling targeted improvements for specific areas of interest. An example of one of the filters we used to parse data from LeanDojo can be found in Figure 4. While these parsed informalizations are more rudimentary compared to our other methods, they allowed us to generate a much larger corpus at minimal cost. Additionally, this approach can be easily expanded upon and provides greater explainability than other methods. However, given the complexity of Lean code, this approach is inherently limited and can only produce rudimentary informalizations such as the ones in Appendix B. We hypothesized that while the model fine-tuned on Regex-parsed statements would enhance performance compared to the baseline, the simplicity

of the translated statements would prevent them from outperforming more sophisticated backtranslation methods. This was confirmed by the results shown in Table 2.

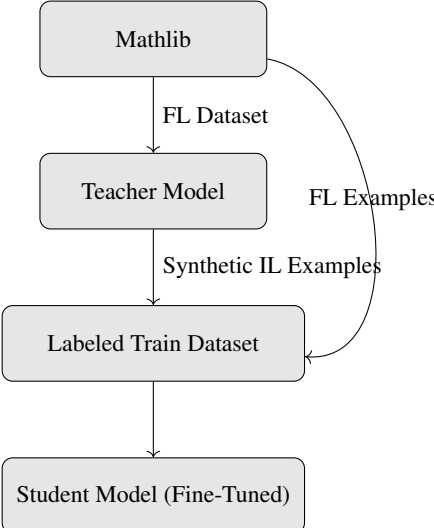

Figure 3: **Distilled Backtranslation Process Graph**. Leveraging the MathLib and ProofNet datasets, a "Teacher" model translates formal theorems into synthetic informal language (IL) examples, which are then backtranslated to augment a training dataset with new formal language (FL) examples. A "Student" model, more compact than the teacher, is fine-tuned on this enriched dataset to improve its translation from IL to FL.

| **Regular Expressions** |
|---|
| ```patterns = {``` 
 "induction": r"induction .+ with .+", 
 "apply": r"apply .+", 
 "rewrite": r"rw .+", 
 "reflexivity": r"refl", 
 "cases": r"cases .+", 
 "introduce": r"intro .+—intros .+", 
 "simplification": r"simp .+", 
 "contradiction": r"contradiction", 
 "exact": r"exact .+", 
 "definition": r"def .+ := .+" } |

Figure 4: **Regular Expressions can be used to generate a large volume of rudimentary informalizations**. Certain Lean proof statements follow predictable patterns that can be captured via a regex. More sophisticated pattern-matching solutions can be built on this framework to increase the amount of paired data available to fine-tune on.

## 3 AI4MATH DATASET OVERVIEW

The integration of data collected through these methodologies resulted in the **AI4Math** Dataset, which encompasses a broad spectrum of proof tactics and informal statements. The total number of tokens from each collection method is provided in Table 1. The following datasets are labeled according to the composition of their contents:

1. *MMA Train* is a large, multilingual, and multi-domain baseline dataset (Jiang et al., 2023) which we aim to compare our generated (IL, FL) pairs against.

2. *On-the-Fly Backtranslation* is comprised of formal-informal pairs created as a result of the backtranslation procedure shown in Figure 1.

3. *GPT-4 MathLib4 (Full Proof)* employs regular expressions to parse the MathLib4 dataset into individual theorems, using GPT-4 with few-shot prompting to informalize a selected subset. Following ProofNet's approach, we applied a 6-shot prompting technique to convert entire theorem statements and their proofs into natural language. This process is shown in Figure 3.

4. *GPT-4 LeanDojo (Individual Tactics)* not only informalizes theorem statements but also each specific tactic within proofs from the LeanDojo dataset, capturing the proof's state before and after each tactic. Using a 0-shot prompting strategy, we created a dataset with informal translations of each tactic, alongside their respective before and after states.

5. *Regex-Parsed LeanDojo Proofs* uses regular expressions to parse specific tactics from the LeanDojo proofs for targeted model refinement. The filters used can be found in Appendix B.

| Data Collection Method | Token Count |
|---|---|
| MMA Train | 10,916,097 |
| Regex-Parsed LeanDojo Proofs | 124,782 |
| GPT-4 MathLib4 (Full Proof) | 71,550 |
| GPT-4 LeanDojo (Individual Tactics) | 1,754 |

Table 1: **Token counts across dataset generation methods**. For each dataset generation method, we compile formal and corresponding generated informal statement data into pairings. This illustrates the diversity and scale of training data available in AI4Math for fine-tuning, as well as demonstrating the relative cost of each synthetic dataset generation methodology.

```
At the end of this explanation, I will give you 2 things. The first is a list of tuples that are
the translations of entire proofs written in Lean, which we will denote the formal language,
to plain English, also known as natural language, as tuples or pairs. This is not an exhaustive
list, these are just examples of informalizations. I will then have a proof written in Lean
represented as a string following the newline character after the list of pairs. Give me the
tuple pair of the proof I give you written in Lean and what you think their natural language
equivalent is given your knowledge of Lean, formatted using LaTeX. Do not output anything
else, just the python tuple I requested. In your output match the exact format "('formal',
'informal')" \n

[("Lean theorem statement 1", "Theorem 1 in natural language"),
("Lean theorem statement 2", "Theorem 2 in natural language"),
("Lean theorem statement 3", "Theorem 3 in natural language"),
("Lean theorem statement 4", "Theorem 4 in natural language"),
("Lean theorem statement 5", "Theorem 5 in natural language"),
("Lean theorem statement 6", "Theorem 6 in natural language")]
```

Figure 5: **Formal-informal pairs used for training are generated via few-shot prompting and distilled backtranslation**. Mined proofs from mathlib are used as input into this format string for prompting, and then GPT-4 creates examples that can be utilized for fine-tuning. In contrast to MMA, our method used six examples to informalize theorem statements.

# 4 AUTOFORMALIZATION PERFORMANCE

To quantify the results of our methods, we evaluate our fine-tuned models for accuracy on ProofNet's test dataset, with the goal of measuring the performance gain over their initial pretrained states. All of our models used GPT-2 as the base architecture and fine-tuned for 3 epochs using a learning rate of 1e-5 and Adafactor optimization, as these hyperparameters for the training setup appeared to yield the best performance. We chose GPT-2 for our experiments due to its open-source availability and lower fine-tuning costs compared to larger, closed-source models like GPT-4. As discussed, future research should consider using larger models to improve outcomes subject to resource availability.

| Fine-Tuning Dataset | Eval Loss |
|---|---|
| GPT-2 Baseline | 75.1839 |
| GPT-4 LeanDojo (Individual Tactics) | 5.1287 |
| Regex-Parsed LeanDojo Proofs | 4.4709 |
| On-the-Fly Backtranslation | 4.032 |
| MMA Train | 3.8791 |
| GPT-4 MathLib4 (Full Proof) | 3.1209 |

Table 2: **Evaluation loss on ProofNet's Test Dataset**. The model used was GPT-2, and it was fine-tuned on each dataset for 3 epochs before being evaluated. GPT-4 MathLib4 (Full Proof) demonstrated higher token efficiency than MMA train, and despite the lower token count, GPT-4 LeanDojo (Individual Tactics) improved the baseline substantially.

We included the results of fine-tuning on the training set of MMA, a dataset that also uses GPT-4 to create the formal-informal pairs that matches the approach we took with the full-proof informalizations (Jiang et al., 2023). However, our methodology uses a 6-shot prompt only, which appears to improve token efficiency as evidenced by the lower token count in Table 1 and lower resulting evaluation loss in Table 2.

We have three major observations:

- **Few-Shot Beats MMA**: Our *GPT-4 MathLib4 (Full Proof)* dataset, developed through a hard-coded 6-shot prompting technique, outperformed the larger, diverse dataset from Albert et al., despite being over 150 times smaller. This substantial increase in token efficiency is likely due to better informalizations of our methodology, which improves upon the comparatively simple prompting technique of MMA. We demonstrate that more examples, though less cost-effective, can result in better autoformalization performance with fewer tokens.

- **Individual Tactics are Efficient**: The *GPT-4 LeanDojo (Individual Tactics)* dataset, which leverages an intensive approach of matching individual tactics to lines in the informalized proof, demonstrated dramatic improvement from GPT-2 baseline with far fewer tokens compared to other methods. However, this method first requires having GPT-4 informalize the proof as a whole, and then pairing the individual tactics to a line in the translated natural language proof: making the strategy more costly at \$0.15 per proof compared to \$0.05 per proof in *GPT-4 MathLib4 (Full Proof)* and the \$0.01 per proof in MMA, which had 332,000 theorems informalized for \$3,500 total (Jiang et al., 2023). This was the largest barrier preventing us from generating a larger training corpus. Given the drastic improvement compared to the baseline, we hypothesize that this is method has significant potential and should be explored in the future with more resources.

- **On-The-Fly Backtranslation Performance Plateau**: When generating the distilled backtranslation dataset for the individual tactics, a review of the data showed that the model trained on our On-the-Fly Backtranslation dataset considerably outperformed the GPT-2 baseline, but its performance plateaued below that of the MMA dataset. We hypothesize that this is due to model size, as both of our best performing methodologies used GPT-4 to generate data for backtranslation.

**Takeaway #1: Prioritize Quality over Quantity!**
Utilizing few-shot prompting techniques appears to generate more accurate and higher-quality parallel informalizations, as evidenced by the results of our *GPT-4 MathLib4 (Full Proof)* and *GPT-4 Lean (Individual Tactics)* datasets. Investing in more sophisticated full-proof few-shot or line-by-line matched examples can significantly enhance token efficiency, yielding better results even with fewer tokens.

**Takeaway #2: Intermediate Steps Matter**
*GPT-4 Lean (Individual Tactics)* demonstrates remarkable proof-efficiency for the small token count despite the higher costs and poor performance compared to utilizing the full proof.

**Takeaway #3: High-Quality Data Beats Diversity.**
Datasets that utilize comparatively richer prompts with GPT-4 exhibit more efficient parallel corpora for autoformalization training compared to datasets that rely on blindly training on diverse data (On-The-Fly Backtranslation, MMA, Regex-Parsed LeanDojo Proofs).

## 5 RELATED WORK

The Multilingual Mathematical Autoformalization (MMA) dataset also uses powerful models GPT-4 to translate from formal language (FL) to informal language (IL) via zero-shot instruction prompting on a large, diverse corpus of mathlib4 data (Jiang et al., 2023). Our approach enhances the efficiency of this process by applying a more intense six-shot prompting strategy on mathlib4 statements; the fine-tuning performance of the resulting dataset not only surpasses that of the MMA dataset on the ProofNet benchmark but does so using only 1/150th of the tokens: significantly optimizing resource use while enhancing output quality. Utilizing a regex, we parsed Lean files from mathlib into over 100,000 tactic proof scripts, and we call this methodology using the "Full-Proof" in pregenerating informal-formal pairs for informalization via GPT-4. Despite the high number of parsed scripts, at an estimated $0.05 per informalization we were only able to informalize a subset of this collection.

Our dataset also utilizes LeanDojo, an open-source platform that offers a comprehensive dataset of over 90,000 Lean theorems. Existing models like ReProver, a retrieval-augmented language model, have demonstrated superior performance compared to established models such as Tidy and GPT-4 (Yang et al., 2023). However, while retrieval-augmented models like ReProver are promising in leveraging formal corpora to enhance LLMs, they rely heavily on automated theorem provers. These models often skip over the detailed intermediate steps that are critical for understanding complex mathematical logic, thereby restricting the systems' flexibility and explainability.

Moreover, the prevalent use of first-order logic in these models constrains their capacity to articulate more sophisticated proofs. Our approach in *GPT-4 LeanDojo (Individual Tactics)* addresses this limitation by directly modeling the incremental deduction characteristic of interactive theorem provers. While this approach involves higher computational costs, Table 2 demonstrates its effectiveness in significantly enhancing baseline performance. Our method achieves notable improvements over the baseline with a very small number of tokens.

## 6 DISCUSSION

Our paper introduces a novel dataset that can be used for enhancing the autoformalization capabilities of large language models. We demonstrate evidence that models can attain superior performance on autoformalization benchmarks such as ProofNet by training on **AI4Math**, and our generation strategies can significantly reduce the number of tokens required compared to MMA.

We also note several limitations with our method. Firstly, our fine-tuning was exclusively conducted using GPT-2. In future research iterations, we anticipate that employing more powerful models — such as Mistral 7B or Llemma 7B for fine-tuning could lead to even better performance. Our "On-The-Fly" backtranslation process was especially constrained by the smaller model size used, and using more powerful models for fine-tuning in future iterations could enhance the quality of informalizations.

Cost constraints were another factor that limited our dataset's expansion, as more sophisticated prompt engineering with GPT-4, particularly for translating and segmenting entire proofs, proved costly. Despite the higher expense, these backtranslation methods surpassed baseline performance with fewer tokens, indicating superior data quality. Given its potential, this approach merits further exploration if sufficient funding can support the additional costs.

# 7 FUTURE WORK

In our current research, we concentrated on autoformalizing theorem statements rather than generating complete formal proofs. This focus was primarily driven by our goal to streamline the process of translating natural language into formal language theorems, which are integral to the groundwork of interactive theorem proving (ITP). However, a significant next step in our exploration involves assessing whether the entire proofs or tactic scripts, when generated by our model, can be successfully compiled within the Lean4 environment.

Additionally, leveraging Interactive Theorem Provers (ITPs) to extract datasets of intermediate proof steps presents a novel opportunity for advancing the task of autoformalization. By sequentially informalizing intermediate statements within a proof, we can generate a dataset that captures the individual steps undertaken by ITPs. This data can subsequently be used to train an LLM, effectively replicating the sequential reasoning performed by ITPs. Since this approach looks to model the incremental deduction used by interactive theorem provers directly, we predict that this could enable language models to develop richer capabilities for reasoning tasks and enable generalization beyond their training proofs.

The ability for a model to generate not just the theorem statements but also the accompanying proofs would mark a substantial leap forward in automating the theorem proving process. It would not only enhance the efficiency and accessibility of formal verification but also potentially transform how complex mathematical and logical reasoning is conducted in computational settings. Thus, the next phase of our research will be dedicated to testing the compatibility and correctness of model-generated proofs with Lean4, aiming to bridge the gap between automated theorem generation and its practical application in ITP systems. This endeavor will necessitate a deeper integration of our model's outputs with the stringent syntax and logical framework of Lean4, posing a challenging yet essential milestone towards fully automated theorem proving.

# 8 CONCLUSION

Our paper has several larger implications for the field of autoformalization. By focusing on autoformalizing theorem statements, we streamline the process of translating natural language into formal language, which is critical for the groundwork of interactive theorem proving. Successfully training an agent for precise autoformalization could drastically reduce the resources needed to convert extensive natural-language mathematical proofs into formal language. This progress would enhance the accessibility and efficiency of automated theorem proving and potentially quicken advancements in related fields like formal verification and program synthesis. Looking forward, a skilled autoformalizer could make all mathematical knowledge, currently documented primarily in natural language, programmable —— significantly broadening the usability of interactive theorem proving systems and accelerating the growth of human mathematical knowledge.

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

## A  APPENDIX A: "FULL-PROOF" PROMPT

We used the following prompt for informalizing the MathLib4 theorems we extracted, with a six-shot example. A complete example can be found in Figure 7.

## B  APPENDIX B: REGEX-PARSED LEANDOJO PROOFS

We used the regular expressions outlined in Figure 4 to parse specific patterns from Lean code. This enabled us to format the parsed strings using the templates in Figure 6 to create a basic 'informalization' of the proofs. This is a tradeoff of quality for quantity, as the informalizations are rudimentary but simple to create. Given the improvements over baseline performance, more complex regex-based systems could contribute to increasing the volume of paired data.

| **Informalization** |
| --- |
| "induction": "We are beginning a proof by induction on {variable}." |
| "apply": "Here, we apply the theorem/lemma {theorem_name}." |
| "rewrite": "We're rewriting part of the expression using {equality_statement}." |
| "reflexivity": "This step concludes that both sides of our equation are identical." |
| "cases": "We're breaking down the problem into cases based on {variable_or_condition}." |
| "introduce": "We introduce new variables {variable_names}." |
| "simplification": "We simplify the current expression or goal using the simp tactic." |
| "contradiction": "This step shows that our assumptions lead to a contradiction." |
| "exact": "Here, we provide the exact term {term_name} that solves our current goal." |
| "definition": "We define a function {function_name} that takes {parameters}." |

Figure 6: **Informalization**. These strings format the parsed patterns into informal explanations.

## C  APPENDIX C: DATA APPENDIX

All of the models and datasets used for fine-tuning can be found on HuggingFace under the AI4M Organization.

At the end of this explanation, I will give you 2 things. The first is a list of tuples that are the translations of entire proofs written in Lean, which we will denote the formal language, to plain English, also known as natural language, as tuples or pairs. This is not an exhaustive list, these are just examples of informalizations. I will then have a proof written in Lean represented as a string following the newline character after the list of pairs. Give me the tuple pair of the proof I give you written in Lean and what you think their natural language equivalent is given your knowledge of Lean, formatted using LaTeX. Do not output anything else, just the python tuple I requested. In your output match the exact format "('formal', 'informal')" \n
[("theorem exists_le_sylow {p : N} {G : Type*} [group G] {P : subgroup G} \n (hP : is_p_group p P) : \n there exists (Q : sylow p G), P <= Q :=", "Let P be a p-subgroup of G. Then P is contained in a Sylow p-subgroup of G."),
("theorem exists_eq_const_of_bounded {E : Type u} [normed_group E] \n [normed_space C E] {F : Type v} [normed_group F] [normed_space C F] \n {f : E to F} (hf : differentiable C f) (hb : metric.bounded (set.range f)) : \n there exists (c : F), f = function.const E c :=", "Let E and F be complex normed spaces and let f:E to F. If f is differentiable and bounded, then f is constant."),
("theorem subset_of_open_subset_is_open (X : Type*) [topological_space X] \n (A : set X) (hA : for every x in A, there exists U: set X, is_open U and x in U and U subset A): \n is_open A :=", "Let X be a topological space; let A be a subset of X. Suppose that for each x in A there is an open set U containing x that U is a subset of A. Then A is open in X."),
("theorem is_multiplicative.eq_iff_eq_on_prime_powers {R : Type*} \n [comm_monoid_with_zero R] (f : nat.arithmetic_function R) \n (hf : f.is_multiplicative) (g : nat.arithmetic_function R) \n (hg : g.is_multiplicative) : \n f = g iff for all (p i : N), nat.prime p implies f (p ^ i) = g (p ^ i) :=", "Two multiplicative functions f, g: N to R are equal if and only if $f(p^i)=f(g^i)$ for all primes p."),
("theorem abs_sum_leq_sum_abs (n : N) (f : N to C) : \n abs (sum i in finset.range n, f i) <= sum i in finset.range n, abs(f i) :=", "If z1, dots, zn are complex, then abs(z1 + z2 + dots + zn) <= abs(z1) + abs(z2) + dots + abs(zn)."),
("theorem distinct_powers_of_infinite_order_element (G : Type*) [group G] (x : G) \n (hx_inf : for all n : N, x ^ n != 1) : \n for all m n : Z, m != n implies x ^ m != x ^ n :=", "If x is an element of infinite order in G, prove that the elements x^n, n in Z are all distinct.")]

Figure 7: **Full Example of our 6-Shot Prompting Methodology.** Such prompts are given to GPT-4 and used to create the formal-informal pairings in our GPT-4 MathLib4 (Full Proof) dataset. Note that characters are translated from Unicode to ASCII for display purposes.

