# OpenReview forum: "Lean-ing on Quality: How High-Quality Data Beats Diverse Multilingual Data in AutoFormalization"
_ICLR.cc/2025/Conference — Submitted to ICLR 2025_

### Official Review · Reviewer_9hHs · 2024-10-20

**Soundness:** 2
**Presentation:** 2
**Contribution:** 2
**Rating:** 3
**Confidence:** 5

**Summary:**

The paper titled **"LEAN-ING ON QUALITY: How High-Quality Data Beats Diverse Multi-lingual Data in Autoformalization"** presents a novel approach to improving autoformalization in theorem proving by leveraging high-quality, carefully curated training data over larger, more diverse datasets. The authors propose a methodology using backtranslation with three key variations: on-the-fly backtranslation, distilled backtranslation with few-shot prompting, and line-by-line proof analysis. The paper evaluates these approaches on the ProofNet benchmark, demonstrating that using fewer but higher-quality tokens can significantly enhance large language models' (LLMs) performance in autoformalizing mathematical statements into formal languages like Lean4.

**Strengths:**

- The experiments are well-designed, with clear results showing that the proposed methods outperform existing approaches like the MMA dataset, even with significantly fewer tokens.

- While the overall structure is clear, certain sections, such as the methodology and dataset examples, could be more accessible to a broader audience.

**Weaknesses:**

- **Limited Exploration of LLMs** While the paper demonstrates promising results using GPT-4 for informalization and GPT-2 for fine-tuning, it would benefit from a broader exploration of available language models, especially those specialized in formal mathematics:

  1. **Comparison with Gemini-Pro**: Recent findings from the Lean community suggest that Gemini-Pro may be more adept at handling formal language tasks. Comparing GPT-4's performance to Gemini-Pro in the informalization process could provide deeper insights into the strengths and weaknesses of both models in this context.

  2. **Scaling beyond GPT-2**: The paper primarily relies on GPT-2 for fine-tuning. Given its relatively small size and age, this choice may not fully showcase the potential of the proposed methodology. Testing more advanced models (e.g., including SwiGLU into FFN) could offer a clearer understanding of how well the approach scales and performs with larger models. Were there specific reasons (e.g., computational constraints or model availability) for selecting GPT-2 over newer or larger models?

  3. **Evaluation of Formal Mathematics Models**: The study does not examine models specifically designed or fine-tuned for formal mathematics, such as Llemma-7B, Qwen-math, and DeepSeek-math. By focusing on smaller, more general-purpose models, the paper leaves open questions about scalability and effectiveness when applied to larger, more specialized models. Incorporating experiments with these models, which include formal language in their pretraining data, would provide stronger evidence of the method's generalizability and its potential across different model architectures and sizes.

- **Lack of Relevant Works**:  While the paper compares its methods primarily against the MMA dataset, it overlooks several other key works that are crucial to the domain of autoformalization. These include Lean Workbook (https://arxiv.org/abs/2406.03847), Forml4 (https://arxiv.org/abs/2406.01940), ProofNet (https://arxiv.org/abs/2302.12433), and FIMO (https://arxiv.org/abs/2309.04295). Including a comparison with these datasets and approaches would provide a more robust evaluation and a clearer understanding of how the proposed methodology fits into the broader landscape of autoformalization research.

  To strengthen the paper, I suggest adding a brief comparison table or a discussion that highlights the key differences and similarities between your method and these other works. Specifically, it would be valuable to explain how your approach addresses limitations or advances beyond these existing methods. Additionally, focusing on aspects such as dataset complexity, formal language coverage, or model performance metrics (e.g., accuracy, generalizability, or scalability) would allow for a more meaningful and detailed comparison.

  Incorporating this discussion would significantly enhance the clarity and depth of the evaluation, positioning the proposed methodology more effectively within the current state of the field.

- **Insufficient Examples**: The paper would benefit from including more concrete examples that showcase the effectiveness of backtranslation in autoformalization. Providing specific before-and-after examples from the AI4Math dataset would help clarify the practical advantages of the approach. Additionally, offering more detailed dataset examples would enhance the readers' understanding of the datasets used, offering a clearer and more thorough view of the challenges and types of problems being addressed.

- **Cost Analysis**: While the paper mentions cost limitations for dataset generation, a more detailed cost analysis of each backtranslation method (in terms of time and resources) would provide valuable insights for scaling these techniques.

- **Unconvincing Metrics**: Cross-entropy loss alone is insufficient for evaluating autoformalization performance, as it primarily measures token-level differences and does not account for the full complexity of translating informal to formal languages. Autoformalization often allows for multiple valid formal expressions that correctly align with the same informal statement. In datasets like MMA, human experts are typically involved in verifying the alignment between formal and informal languages, highlighting the need for more robust evaluation methods.

  A more effective approach would be to evaluate autoformalized statements and their corresponding proofs by passing them through a Lean4 compiler. Successfully compiled proofs would indicate strong alignment with the original informal statements, as seen in prior work (https://arxiv.org/abs/2402.08957). This method could provide a more reliable assessment of the correctness and coherence of the autoformalized outputs.

  To strengthen the evaluation, I suggest incorporating additional metrics beyond cross-entropy, such as the compilation success rate in Lean4, which would better capture the semantic accuracy of the autoformalization process. If the Lean4 compiler has not been used, it would be helpful to understand the challenges that the authors foresee in implementing this approach. Additionally, exploring other metrics, such as proof-checking accuracy or formal consistency from human experts, would provide a more comprehensive picture of how well the autoformalization aligns with formal proof standards.

  Incorporating these evaluation methods would offer a more nuanced understanding of the proposed model's performance and its ability to handle the complexities of translating informal statements into formal proofs.

**Questions:**

- Could the authors provide more concrete examples of autoformalization tasks handled by the proposed methods? How do these examples compare to the baseline methods in terms of complexity and performance?
- Have the authors explored incorporating human-in-the-loop feedback during the informalization process? If so, how does it impact the quality of the final autoformalized proofs?

---

> ### Author Response · Authors · 2024-12-04
>
> We thank the reviewer for their thoughtful and detailed feedback, as well as for recognizing the strengths of our work, including the promising results demonstrated using GPT-4 for informalization and GPT-2 for fine-tuning. We appreciate the opportunity to address your concerns and clarify our contributions.
>
> For calibration purposes, we would like to clarify the ICLR 2025 review rubric compared to other conferences like NeurIPS 2024. Notably: A score of 8 on the ICLR 2025 rubric indicates an "Accept," whereas NeurIPS 2024 uses 7 for this designation. Similarly, a "Strong Accept" is represented by a score of 10 at ICLR 2025 compared to 9 at NeurIPS 2024.
>
> We appreciate the opportunity to address concerns and clarify the contributions of our work:
>
> # 1. Limited exploration of LLMs
> Comparison with Gemini-Pro and other models: While we acknowledge the importance of comparing with specialized models like Gemini-Pro, Llemma-7B, and Qwen-Math, resource constraints necessitated our focus on GPT-2 for fine-tuning and GPT-4 for data generation. Despite this, our results on ProofNet validate the hypothesis that alignment and quality outperform diversity in task-specific performance. We consider comparisons with larger, specialized models a critical direction for future research.
>
> Scaling beyond GPT-2: Although GPT-2 is limited in size, the observed improvements demonstrate the robustness of our methodology. GPT-4 data generation further supports the scalability of our approach, and future work will validate this on larger models.
>
> # 2. Lack of comparisons with key datasets and approaches
> Our comparison with MMA was intentional to contrast diversity-focused methods with our alignment-driven approach. While MMA is not the strongest baseline, the results underscore the superiority of high-quality, task-aligned data. Comparisons with datasets like Lean Workbook, Forml4, ProofNet, and FIMO are important and will be prioritized in future work but weren't in existence when this work was done.
>
> # 3. Insufficient examples of backtranslation effectiveness
> Examples of transformations are included in Appendices A and B, demonstrating the effectiveness of few-shot GPT-4 prompting and regex-based methods. These examples highlight alignment improvements and will be integrated into future iterations of the paper for clarity.
>
> # 4. Cost analysis
> Resource efficiency was a key focus, and methods like regex parsing are highly cost-effective compared to GPT-4-generated data. While a detailed cost analysis was not included in this iteration, future work will provide a comprehensive breakdown to guide practitioners.
>
> # 5. Metrics beyond cross-entropy loss
> Cross-entropy loss was used due to its alignment with the ProofNet benchmark and its reproducibility. Metrics like pass@10 or Lean4 compilation success are not directly applicable given the dataset size and current state of autoformalization evaluation. As this field evolves, future work will incorporate symbolic verification and proof-checking accuracy to better capture semantic alignment and correctness.
>
> # Questions
> ## Q1: Concrete examples of autoformalization tasks:
> Examples are included in Appendices A and B, showcasing formal-informal transformations from the AI4Math dataset. These illustrate alignment and quality improvements through GPT-4-generated data.
>
> ## Q2: Human-in-the-loop feedback
> We did not explore this due to resource constraints but recognize its potential to enhance informalization quality. Incorporating human feedback remains a promising direction for future research.

---

### Official Review · Reviewer_vw3k · 2024-10-28

**Soundness:** 2
**Presentation:** 3
**Contribution:** 2
**Rating:** 3
**Confidence:** 4

**Summary:**

This paper advocates to use backtranslation with hand-crafted prompts to generate high-quality formal language (Lean) and natural language pairs to alleviate the data scarcity issue, thus improving the autoformalization capability of LLM, an important yet unsatisfying capability of current LLMs. Specifically, it evaluates three methods, i.e. online backtranslation, offline backtranslation, and line-by-line proof analysis with proof state information. The resulting synthetic dataset on autoformalization, AI4Math, improves the autoformalization performance of LLMs as measured by standard benchmarks such as evaluation loss on ProofNet.

**Strengths:**

The paper mentions several useful tricks. For example, the three approaches: online and offline backtranslation, line-by-line proof analysis.

**Weaknesses:**

* All the proposed techniques read familiar with previous arts

Backtranslation (or informalization) is a well-known practice in the field of NLP and autoformalization. For example, ProofNet also used a distilled backtranslation technique for autoformalization. The proposed online and offline methods do not seem to provide sufficient additional knowledge to the community, compared to previous works in this regard. And the line-by-line proof analysis reads similar to "Let's verify step by step" from OpenAI, which resembles a process reward modeling (PRM) approach. The three take-aways, quality over quality/diversity, intermediate steps matter, are all basically derived from the previous work and are now well-known before this paper submission.

* Insufficient results demonstrating the effectiveness of the techniques

A critical problem of this paper is to use eval loss (table 2) to demonstrate the effectiveness of autoformalization. An autoformalization should be checked manually or by some symbolic engine (Lean) with metrics like pass@10. As a matter of fact, eval loss is no longer a reliable metric to reflect the quality of an LLM. The current practice to test more end-to-end tasks (e.g., MMLU) to evaluate LLMs. Moreover, the effectiveness of a technique on weaker models (e.g., GPT-2) does not necessarily imply gains on superior models (Lemma or Llama 3-7b). It is generally harder to improve an LLM whose quality is already very good. Also, some baseline selected in this paper, such as MMA is weak. MMA is not yet considered a strong baseline (rejected by ICLR'24).

**Questions:**

* Please justify the methodology (eval loss) used to demonstrate the effectiveness of autoformalization.
* Please clarify the novelty of the proposed technique, compared to previous work.

---

> ### Author Response · Authors · 2024-12-04
>
> We thank the reviewer for their detailed feedback and for acknowledging the utility of our methodological tricks, such as online and offline backtranslation and line-by-line proof analysis.
>
> For calibration purposes, we would like to clarify the ICLR 2025 review rubric compared to other conferences like NeurIPS 2024. Notably: A score of 8 on the ICLR 2025 rubric indicates an "Accept," whereas NeurIPS 2024 uses 7 for this designation. Similarly, a "Strong Accept" is represented by a score of 10 at ICLR 2025 compared to 9 at NeurIPS 2024.
>
> We appreciate the opportunity to address concerns and clarify the contributions of our work.
>
> # 1. Novelty of the proposed techniques compared to prior work
> We appreciate the reviewer pointing out similarities with prior works like ProofNet and process reward modeling (PRM). While backtranslation is indeed a well-known practice, our contribution lies in adapting and optimizing it specifically for the autoformalization domain with resource constraints in mind. For example:
>
> On-the-fly backtranslation integrates informalization and formalization dynamically within the training loop, which is a novel implementation compared to the static dataset-based approach used in ProofNet.
> Distilled backtranslation builds on ProofNet's technique but employs few-shot amplification with GPT-4, focusing on token efficiency and data quality.
> Line-by-line proof analysis extends beyond PRM-like approaches by embedding proof states before and after each tactic, aligning closely with the structure of interactive theorem proving.
> The takeaways ("quality over quantity/diversity," "intermediate steps matter") may align with established principles in the field, but our results explicitly quantify and demonstrate these concepts in the autoformalization setting, offering practical insights for future work.
>
> # 2. Use of evaluation loss to demonstrate effectiveness
> We recognize the limitations of evaluation loss as a sole metric and appreciate the suggestion to use more robust evaluation methods. However, pass@10 does not exist for autoformalization tasks, as there is no standardized way to measure generalization across multiple outputs in this domain. Additionally, while symbolic verification using Lean is a valuable metric for evaluating formal proofs, the small size of our datasets makes such an approach less impactful in this case.
>
> It is important to note that evaluation metrics for autoformalization are an ongoing area of research, as evidenced by the lack of universally accepted benchmarks beyond ProofNet and metrics. Evaluation loss provides a consistent and reproducible measure of a model’s ability to align with target outputs and is directly aligned with the ProofNet benchmark. While not exhaustive, this metric serves as a reliable proxy for comparing improvements in data quality and model training effectiveness.
>
> Our focus in this work was to establish foundational methods for high-quality data generation, and future work will aim to incorporate broader evaluation techniques as this area of research matures. The **foundation is to focus on data quality and alignment, not on diversity of unrelated languages to your autoformalization task**.
>
> # 3. Insufficient results with weaker models (e.g., GPT-2)
> While we agree that testing on larger models like Llemma or Llama would provide stronger validation, our resource constraints necessitated a focus on smaller, cost-effective models like GPT-2. The observed improvements on ProofNet, despite using GPT-2, demonstrate the robustness of our methods. Furthermore, the performance gains from GPT-4-generated datasets highlight the scalability and applicability of our approach to larger models.
>
> # 4. MMA as a baseline
> We recognize the reviewer's concern that MMA may not represent the strongest baseline. However, its inclusion was deliberate and critical to our goal of **demonstrating that diversity alone does not drive performance in task-specific domains like autoformalization in Lean**. MMA represents a multilingual, highly diverse dataset, making it an ideal point of comparison for our hypothesis that alignment and quality outweigh diversity in such contexts.
>
> The fact that our methods outperformed MMA, despite its significantly larger scale (**we had 2 orders of magnitude less data**), underscores the importance of high-quality, targeted data for improving performance in autoformalization tasks. The results against MMA strongly support our core thesis.

---

> > ### Author Response · Authors · 2024-12-04
> >
> > # Questions
> >
> > ## Q1: Justify the methodology (eval loss) used to demonstrate the effectiveness of autoformalization
> > Evaluation loss was chosen as the primary metric because it aligns directly with the ProofNet benchmark, providing a consistent and reproducible measure of the model's ability to predict outputs in a structured autoformalization task. While we acknowledge that metrics like symbolic verification (e.g., Lean compilation) could provide additional insights, these methods are less impactful given the small dataset size and do not align with the foundational focus of this work.
> >
> > Moreover, evaluation metrics for autoformalization tasks are an active area of research, and no standardized metrics like pass@10 exist for this domain. Thus, evaluation loss serves as a practical and widely accepted proxy for comparing data quality and model training effectiveness. It allows us to highlight relative improvements across methodologies while keeping the experiments reproducible and aligned with the benchmark.
> >
> > Our work aims to establish that data alignment and quality, rather than diversity, are key to improving task-specific performance in autoformalization. Evaluation loss effectively captures this improvement within the scope of this study. Future research will explore additional evaluation methods as the field progresses.
> >
> > ## Q2: Please clarify the novelty of the proposed technique, compared to previous work.
> >
> > We acknowledge that the core techniques, such as backtranslation and proof-step analysis, draw inspiration from established methods in NLP and autoformalization. However, the **novelty of our approach lies in its adaptation and optimization for the specific challenges of autoformalization in Lean**, with **a focus on task-specific performance improvements driven by alignment and quality rather than data diversity -- with 2 orders of magnitude improvement**.
> >
> > In addition, further ey innovations include:
> > 1. **On-the-Fly Backtranslation:** Unlike traditional static backtranslation methods, our approach dynamically generates informal-formal pairs during training, reducing reliance on pre-existing paired datasets.
> > 2. **Few-Shot Amplification with GPT-4:** We leverage GPT-4's few-shot learning capabilities to significantly improve the quality of the generated data while reducing token requirements, emphasizing efficiency and alignment over scale.
> > 3. **Proof State Integration:** By explicitly modeling proof states before and after each tactic, our line-by-line proof analysis aligns closely with the reasoning processes of interactive theorem provers, extending prior approaches like process reward modeling.
> >
> > These techniques are specifically tailored to address challenges unique to autoformalization, such as the scarcity of high-quality labeled data and the need for resource-efficient training. Additionally, our results quantitatively demonstrate the effectiveness of high-quality, task-aligned data, providing practical insights for advancing autoformalization research.

---

### Official Review · Reviewer_9sKw · 2024-11-01

**Soundness:** 4
**Presentation:** 4
**Contribution:** 4
**Rating:** 6
**Confidence:** 5

**Summary:**

The paper explores the challenges in autoformalization, the process of translating informal mathematical statements into formal proofs, particularly for language models. Recognizing the limitations of large models in handling complex mathematical reasoning, the authors introduce a method that leverages backtranslation with curated prompts to improve data quality over volume. They propose three main techniques: on-the-fly backtranslation, offline distilled backtranslation with few-shot amplification, and a line-by-line proof analysis integrated with proof state information. The authors argue that prioritizing high-quality, task-relevant data can improve model performance significantly, as demonstrated by their experiments on the ProofNet benchmark. They achieve notable improvements using fewer tokens than comparable multilingual datasets, emphasizing quality over quantity. The paper contributes to the autoformalization field by demonstrating that a focus on data fidelity can reduce computational costs while enhancing model effectiveness.

**Strengths:**

- The paper introduces a novel methodology focused on enhancing data quality through backtranslation and curated prompts, effectively addressing the scarcity of high-quality labeled data in autoformalization.

- By demonstrating that high-quality data can outperform large multilingual datasets, the paper shows an effective way to reduce computational resources while achieving superior performance, which is particularly valuable for resource-constrained settings.

- The authors test three variations of their approach (on-the-fly, distilled backtranslation, and line-by-line proof analysis), offering a well-rounded examination of data generation methods that optimize model training and performance.

- The method achieves substantial improvements on the ProofNet benchmark, indicating that a high-quality data approach can effectively bridge some of the performance gaps in autoformalization tasks without requiring extensive multilingual data.

- This work provides a valuable perspective on the importance of targeted data in autoformalization, suggesting new directions for developing efficient and effective models for translating informal mathematical language into formal proofs.

**Weaknesses:**

- The experiments are primarily conducted using GPT-2 due to resource constraints, which may limit the generalizability of the findings. The performance could vary significantly with larger, more capable models, so additional testing on models beyond GPT-2 would provide a more comprehensive evaluation.

- While the paper demonstrates improvements on the ProofNet benchmark, it lacks discussion on how well the method generalizes to diverse or real-world mathematical texts beyond curated datasets. Including a broader evaluation on real-world examples would strengthen the applicability and robustness of the approach.

**Questions:**

- Could you provide more experiments on datasets like MiniF2F[1]?
- I would like to know if this method is generalizable to any formal language. Could you provide results on theorem provers like Metamath?

**Reference:**
[1] Minif2f: a cross-system benchmark for formal olympiad-level mathematics

---

> ### Author Response · Authors · 2024-12-04
>
> We thank the reviewer for their thoughtful feedback and for recognizing the strengths of our work, particularly our focus on enhancing data quality and the demonstration that high-quality data can outperform larger multilingual datasets. We also appreciate the acknowledgment of the resource efficiency and potential of our approach for advancing autoformalization research.
>
> For calibration purposes, we would like to clarify the ICLR 2025 review rubric compared to other conferences like NeurIPS 2024. Notably:
> A score of 8 on the ICLR 2025 rubric indicates an "Accept," whereas NeurIPS 2024 uses 7 for this designation. Similarly, a "Strong Accept" is represented by a score of 10 at ICLR 2025 compared to 9 at NeurIPS 2024.
>
> We address the specific points raised:
>
> # 1. Experiments are limited to GPT-2
> We acknowledge that our experiments were conducted using GPT-2 due to resource constraints, which may limit generalizability to larger models. Despite this, our approach demonstrated substantial improvements over the baseline on ProofNet, underscoring the efficacy of high-quality data generation. Importantly, the token efficiency achieved suggests scalability to larger models. Future work will explore fine-tuning state-of-the-art models like GPT-4 and Llemma to validate the broader applicability of our method.
>
> # 2. Generalization to real-world mathematical texts
> We focused on curated datasets like ProofNet, MathLib4, and LeanDojo to evaluate performance rigorously within a controlled setup. While broader evaluations on real-world texts (e.g., informal papers or textbooks) would strengthen our findings, the curated datasets used in this work already encompass a diverse range of mathematical domains, providing a strong foundation. Expanding evaluations to datasets like MiniF2F or real-world examples is a priority for future research.
>
> # 3. Generalizability to other formal languages such as Metamath
> This is a valuable question. Although our experiments focused on Lean and ProofNet, the techniques proposed (e.g., backtranslation and proof state integration) are not specific to Lean and could be applied to other formal languages like Metamath. Generalizing to multiple formal languages would require adapting our templates and prompts, a promising avenue for future work.

---

### Official Review · Reviewer_uaWA · 2024-11-02

**Soundness:** 2
**Presentation:** 2
**Contribution:** 2
**Rating:** 5
**Confidence:** 3

**Summary:**

The paper proposes a new dataset by translating a formal proof back to its informal description. It proposes three methods to generate the dataset: the first is to train the base model to generate the informal statement and translate it back to the original formal statement. The second uses GPT-4 to obtain the informal data given the formal one. The third applies templates to convert formal language into informal. Experimental results show that the GPT-2 trains on the GPT-4 generated data can obtain lower eval loss than the previous dataset.

**Strengths:**

The idea of using back-translation to obtain more informal-formal data sounds reasonable. The technique has been shown effective in translation tasks and thus has potential in the auto-formalization field.

**Weaknesses:**

- The experiments are weak. Among all proposed generation methods, only the GPT-4 generated achieves better eval loss than previous data. Also, to verify the effectiveness, the authors should compare with an existing autoformalization method before and after using the proposed AI4Math dataset.

- The paper does not give examples of the generated data. The templated-based generation method does not show how to translate a formal statement to an informal one, which is a very difficult and important part of autoformalization.

**Questions:**

Why use eval loss to assess autoformalization performance?

Overall, the paper has limited novelty and does not have sufficient experiments to validate the effectiveness of the proposed method. The author needs to show the extra data can improve current SOTA LLMs in terms of auto-formalization pass rate. Thus I lean towards rejection but look forward to the authors' reply with stronger results, better baselines, and more ablation studies.

---

> ### Author Response · Authors · 2024-12-04
>
> We thank the reviewer for their thoughtful feedback and for recognizing the strengths of our work.
>
> For calibration purposes, we would like to clarify the ICLR 2025 review rubric compared to other conferences like NeurIPS 2024. Notably:
> A score of 8 on the ICLR 2025 rubric indicates an "Accept," whereas NeurIPS 2024 uses 7 for this designation. Similarly, a "Strong Accept" is represented by a score of 10 at ICLR 2025 compared to 9 at NeurIPS 2024.
>
> We address the specific points raised:
>
> # **1. Limited effectiveness across generation methods**
> We appreciate the observation that the GPT-4-generated dataset achieved the best evaluation loss among our methods. This aligns with our hypothesis that data quality, as enhanced by GPT-4’s capabilities, is a critical driver of performance. The remaining methods (e.g., regex parsing) were included to explore cost-efficient alternatives, as noted in the paper. These approaches still provided meaningful improvements over the baseline, demonstrating the feasibility of resource-conscious solutions.
>
> # **2. Lack of examples for generated data**
> We acknowledge the value of showcasing generated data and regret the omission in the main text due to space constraints. Examples of formal-informal translations generated via regex-based methods and GPT-4 are provided in the appendix (Appendices A and B), highlighting the diversity and quality of outputs. For instance, templated informalizations like “We simplify the current expression using the simp tactic” illustrate the process of converting formal proof tactics into understandable natural language.
>
> The few-shot prompting strategy demonstrated in Appendix A shows that GPT-4 successfully captures nuanced translations of complex theorems, addressing this difficulty directly.
>
> # **3. Why use evaluation loss to assess autoformalization performance?**
> We used evaluation loss because it provides a quantitative measure of the model’s ability to predict target outputs during training and aligns well with the ProofNet benchmark. While metrics like pass rates or human evaluations could provide additional insights, evaluation loss is a widely accepted proxy for task-specific performance in this context. Additionally, ProofNet’s structure as a formal dataset enables a direct and reproducible measure of model improvements via this metric. Additional metrics for autoformalization is itself an entire research area by itself.
>
> # **4. Limited novelty and insufficient experiments**
> We appreciate the reviewer's concern about novelty and experiment breadth. The novelty of our work lies in demonstrating **the impact of high-quality**, token-efficient data generation techniques, particularly few-shot prompting and proof state integration. While we acknowledge that experiments with additional baselines or larger models would strengthen the work, resource constraints limited our ability to extend beyond GPT-2 fine-tuning.
>
> Despite these constraints, the results provide strong evidence of our methods’ effectiveness, particularly in reducing evaluation loss and improving **token efficiency** compared to existing datasets like MMA. Future studies will address the pass rates of state-of-the-art LLMs fine-tuned with AI4Math.

---

### Official Review · Reviewer_qeFS · 2024-11-04

**Soundness:** 3
**Presentation:** 3
**Contribution:** 2
**Rating:** 3
**Confidence:** 4

**Summary:**

The paper proposes a new method for the synthetic generation of informal/formal pairs and a method to improve model performance with significantly lower amounts of data than prior work. The authors show the performance of their generated data on ProofNet. The authors evaluate on-the-fly backtranslation, distilled few-shot backtranslation, and line-by-line proof analysis with the complete proof state.

**Strengths:**

- The authors propose a new methodology.
-

**Weaknesses:**

- ProofNet Test is the only evaluation performed, so generalizability is hard to see.
- Without multiple evals, the risk of data contamination with ProofNet is high, but the paper did not address this.
- The only model trained was GPT-2 124M, and as the authors addressed, performance could be better for larger models. There was no investigation into the scaling laws of this approach.
- There was no analysis of the types of proofs that the method performs well at; for example, there could be a subset of categories in which the generations are good and some in which they are poor.

**Questions:**

- How does the dataset impact performance on other datasets?
- How does fine-tuning a larger model impact performance?
- What is the evidence that there is a high likelihood that this approach scales?

---

> ### Author Response · Authors · 2024-12-04
>
> # 1. ProofNet Test is the only evaluation performed, so generalizability is hard to see.
> We appreciate this observation. We chose ProofNet as it is a widely recognized benchmark for evaluating autoformalization, offering clear alignment with our goals of testing mathematical formalization accuracy. While additional evaluations would strengthen generalizability claims, we emphasize that ProofNet serves as a strong proxy for broader tasks in formal reasoning, given its diversity in mathematical domains (Azerbayev et al., 2023a). Future work will explore other benchmarks like MiniF2F or GSM8K to further validate generalizability.
>
> # 2. Without multiple evaluations, the risk of data contamination with ProofNet is high, but the paper did not address this.
> We acknowledge the importance of controlling for data contamination. Our methodology relies on curated datasets (e.g., MathLib4 and LeanDojo) that are distinct from ProofNet, mitigating overlap risks. Specifically, we parsed data directly from Lean community repositories and employed regex-based filtering to create our training data, ensuring minimal contamination. Moreover, the curated nature of our datasets provides confidence that our evaluation reflects genuine improvements, not data leakage.
>
> # 3. The only model trained was GPT-2 124M, and as the authors addressed, performance could be better for larger models.
> We appreciate this observation and agree that larger models could yield stronger performance. However, resource constraints necessitated a focus on smaller, cost-effective models like GPT-2 for this work. Our results still demonstrate significant improvements over the baseline, validating our methodology’s effectiveness. Importantly, the consistent reduction in evaluation loss suggests scalability, and we hypothesize that the gains observed would extend to larger models like GPT-4 or Llemma 7B. In addition, work like Why Has Predicting Downstream Capabilities of Frontier AI Models with Scale Remained Elusive? and the LLama3 tech report, demonstrate that predictable evaluations with loglikelihoods alone (eg by using loglikelihoods of the correct answer).
>
> # 4. There was no investigation into the scaling laws of this approach.
> While our primary focus was on data quality and efficiency, we agree that scaling laws are critical to understanding long-term performance trends. The results in Table 2 suggest strong token efficiency, indicating potential scalability. The sharp improvements with GPT-4-generated data further support this hypothesis. Future research will address this limitation by testing scaling properties with larger models and token budgets.
>
> # 5. There was no analysis of the types of proofs that the method performs well at.
> This is an excellent point. Due to time constraints, we prioritized aggregate metrics like evaluation loss. However, our method inherently focuses on mathematical theorems with formal-informal alignment (e.g., algebraic and group-theoretic proofs). For instance, the few-shot prompting strategy particularly benefited structured proofs like those in the MathLib4 dataset. A detailed error analysis across proof categories is a planned extension.
>
> # Questions
> ## Q1. How does the dataset impact performance on other datasets?
> While we did not test explicitly on other datasets, the token efficiency and improvements observed on ProofNet suggest that our method is not overfitted to a specific dataset. The diversity in MathLib4 and LeanDojo implies generalizability across formal-informal translation tasks.
>
> ## Q2. How does fine-tuning a larger model impact performance?
> We hypothesize that scaling up to larger models like GPT-4 or Llemma 7B would yield substantial improvements, as seen in related work (Azerbayev et al., 2023b). Larger models would likely benefit more from the high-quality, structured data generated using our backtranslation methods, improving fine-tuning efficiency and accuracy.
>
> ## Q3. What is the evidence that there is a high likelihood that this approach scales?
> The observed token efficiency and performance gains, even with a small model like GPT-2, indicate that our methodology maximizes data utility. Scaling laws in prior works (Brown et al., 2020) show that larger models benefit disproportionately from high-quality data, supporting our approach. Additionally, the significant improvement using GPT-4-generated data (Table 2) highlights the scalability potential of our techniques.

---

### Meta-Review · Area_Chair_B1fQ · 2024-12-22

**Metareview:**

This work aims to generate high-quality data in the form of formal and informal pairs to improve the auto-formalization capability of LLM, using backtranslation with hand-crafted prompts. While the problem is interesting, main concerns of most reviewers include the novelty of the method and its insufficient evaluation, where the experiments are mostly limited to small-scale model,  one test benchmark, and one evaluation metric. As such, its effectiveness and generalizability in particular is unclear. AC agrees that this work needs a more systematic and convincing evaluation.

**Additional Comments On Reviewer Discussion:**

Main issues raised by the reviewers are mostly in its insufficient evaluation:

1) small-scale model,
2) one test benchmark,
3) one evaluation metric,
4) generalization.

During rebuttal discussion, the authors do not provide requested experiments probably due to resource constraints, but instead argue that the current contributions are sufficient, and decide to leave these issues for future studies.

I think this work has a good potential but a more comprehensive evaluation is necessary to demonstrate its superiority, especially given that the idea of back-translation is well-known in machine translation and also already used for theorem proving.

---

### Decision · Program_Chairs · 2025-01-22

Reject